# Neoadjuvant–adjuvant pertuzumab in HER2-positive early breast cancer: final analysis of the randomized phase III PEONY trial

Liang Huang[1,2], Da Pang[3], Hongjian Yang[4], Wei Li[5], Shusen Wang[6], Shude Cui[7], Ning Liao[8], Yongsheng Wang[9], Chuan Wang[10], Yuan-Ching Chang[11], Hwei-Chung Wang[12], Seok Yun Kang[13], Jae Hong Seo[14], Kunwei Shen[15], Suphawat Laohawiriyakamol [16], Zefei Jiang[17], Haiyan Wang[18], François Lamour[19,22], Grace Song[20], Michelle Curran[19], Chunzhe Duan[21], Sanne Lysbet de Haas[19], Eleonora Restuccia[19] & Zhimin Shao [1,2] ✉

The randomized, multicenter, double-blind, placebo-controlled, phase III PEONY trial (NCT02586025) demonstrated significantly improved total pathologic complete response (primary endpoint) with dual HER2 blockade in HER2-positive early/locally advanced breast cancer, as previously reported. Here, we present the final, long-term efficacy (secondary endpoints: event-free survival, disease-free survival, overall survival) and safety analysis (62.9 months' median follow-up). Patients (female; $n = 329$; randomized 2:1) received neoadjuvant pertuzumab/placebo with trastuzumab and docetaxel, followed by adjuvant 5-fluorouracil, epirubicin, and cyclophosphamide, then pertuzumab/placebo with trastuzumab until disease recurrence or unacceptable toxicity, for up to 1 year. Five-year event-free survival estimates are 84.8% with pertuzumab and 73.7% with placebo (hazard ratio 0.53; 95% confidence interval 0.32–0.89); 5-year disease-free survival rates are 86.0% and 75.0%, respectively (hazard ratio 0.52; 95% confidence interval 0.30–0.88). Safety data are consistent with the known pertuzumab safety profile and generally comparable between arms, except for diarrhea. Limitations include the lack of ado-trastuzumab emtansine as an option for patients with residual disease and the descriptive nature of the secondary, long-term efficacy endpoints. PEONY confirms the positive benefit:risk ratio of neoadjuvant/adjuvant pertuzumab, trastuzumab, and docetaxel treatment in this patient population.

The availability of human epidermal growth factor receptor 2 (HER2)-targeted therapy has dramatically changed the prognosis for patients with HER2-positive early breast cancer. Trastuzumab added to chemotherapy has been associated with improved pathologic complete response (pCR) rates and long-term survival[1–3]. However, the majority of patients do not achieve a pCR[1,4] and of these, almost one third will develop disease recurrence at 3 years[4]. Dual HER2 blockade with pertuzumab and trastuzumab has proven more effective than single-agent trastuzumab in the treatment of HER2-positive early breast cancer. The APHINITY study (NCT01358877) showed that addition of pertuzumab to standard adjuvant therapy significantly improves invasive disease-free survival (DFS)[5–7]. Dual HER2 blockade evidently increases the frequency of diarrhea in HER2-positive breast cancer, but not the rate of cardiac adverse events (AEs)[5,8,9].

NeoSphere (NCT00545688) was the first phase II study to demonstrate improved pCR rates through the addition of pertuzumab to the neoadjuvant trastuzumab–docetaxel treatment regimen in HER2-positive early, locally advanced, or inflammatory breast cancer[10]. Though progression-free survival and DFS after 5 years' follow-up of NeoSphere showed large and overlapping confidence intervals (CIs), the primary endpoint of pCR was still supported[11], suggesting the addition of neoadjuvant pertuzumab is potentially beneficial with respect to long-term outcomes. However, patients did not receive adjuvant pertuzumab.

Consequently, the randomized, multicenter, double-blind, placebo-controlled phase III PEONY trial (NCT02586025) was conducted to compare the efficacy, safety, and tolerability of adding pertuzumab to trastuzumab and docetaxel in Asian patients with HER2-positive early or locally advanced breast cancer. The PEONY trial, to our knowledge, is the first randomized phase III study of the addition of pertuzumab to trastuzumab in the neoadjuvant and adjuvant setting in this population. The PEONY trial met its primary endpoint: the addition of pertuzumab resulted in a statistically significant and clinically meaningful improvement in total pCR (tpCR) rate as assessed by an independent review committee, with tpCR rates similar to those in NeoSphere[10,12]. The results of prespecified sensitivity analyses and subgroup analyses were consistent with the results of the primary endpoint analysis in the intention-to-treat (ITT) population[12].

Here, we report the final analysis of long-term efficacy (at 3 and 5 years) in PEONY, including data from the adjuvant treatment period, overall treatment period, and treatment-free follow-up period, as well as the prespecified secondary endpoints of event-free survival (EFS), DFS, overall survival (OS), and safety. Both EFS and DFS rates at 3 and 5 years are shown to be higher in the pertuzumab arm compared with the placebo arm. There are few OS events in either arm, but a positive trend towards improved OS is seen with pertuzumab compared with placebo. Safety data are in line with the known pertuzumab safety profile and are generally comparable between arms. The magnitude of clinical benefit and acceptable safety data in this study confirm the positive benefit:risk ratio of neoadjuvant/adjuvant pertuzumab, trastuzumab, and docetaxel treatment in patients with HER2-positive early or locally advanced breast cancer.

## Results

The primary outcome has been reported previously[12].

### Patient population

From March 14, 2016, to March 13, 2017, a total of 329 female patients were randomized (pertuzumab arm: 219; placebo arm: 110 [Fig. 1]). A total of 175 (79.9%) and 82 patients (74.5%), respectively, had completed the study with 5 years' follow-up (cutoff: March 14, 2022; median follow-up: 62.9 months).

Baseline demographics and disease characteristics were generally well balanced between treatment arms (Supplementary Table 1). A total of 170 patients (51.7%) had hormone receptor-positive disease and 100 patients (30.4%) belonged to the locally advanced stage.

The breast conserving surgery rates were similar between the pertuzumab and placebo arms (57 [26.0%] versus 35 [31.8%]). A similar trend was also seen with the sentinel lymph node biopsy rates (34 [15.5%] versus 17 [15.5%]). The most common reasons for study discontinuation were withdrawal by subject (pertuzumab: 22 patients [10.0%]; placebo: 8 [7.3%]) and death (pertuzumab: 12 [5.5%]; placebo: 11 [10.0%]). Exposures to the study drugs are shown in Supplementary Tables 2–4.

### Efficacy outcomes

The efficacy-evaluable (ITT) population consisted of 219/219 patients in the pertuzumab arm and 110/110 patients in the placebo arm. Five years after randomization of the last patient, 59 of 329 (17.9%) patients had disease recurrence/progression or had died. The majority of first EFS events (Table 1) were distant metastases without central nervous system metastases, with fewer patients with events in the pertuzumab versus placebo arm (13 [5.9%] versus 12 [10.9%]). However, all six central nervous system metastases as first events occurred in the pertuzumab arm. Locoregional recurrences after surgery occurred more frequently in the placebo versus pertuzumab arm (seven patients [6.4%] versus two [0.9%]).

The 3-year EFS rates were 88.9% in the pertuzumab arm and 79.7% in the placebo arm ($\Delta$9.2%; 95% CI 0.29–18.1; $p = 0.043$ [Fig. 2a]). The 5-year EFS rates were 84.8% and 73.7% ($\Delta$11.1%; 95% CI 1.2–21.0; $p = 0.027$), respectively, with a hazard ratio (HR) of 0.53 (95% CI 0.32–0.89 [Fig. 2a]).

Post-surgery, the 3-year DFS rate was 90.1% in the pertuzumab arm and 81.1% in the placebo arm ($\Delta$9.0%; 95% CI 0.30–17.7; $p = 0.043$ [Fig. 2b]). The 5-year DFS rates were 86.0% and 75.0%, respectively ($\Delta$11.0%; 95% CI 1.2–20.7; $p = 0.028$) with a HR of 0.52 (95% CI 0.30–0.88 [Fig. 2b]).

OS events were low in both arms; 12 (5.5%) deaths occurred in the pertuzumab arm and 11 (10.0%) in the placebo arm. Twenty-three deaths occurred overall: 15 due to progressive disease or disease recurrence and two due to sudden death (Supplementary Table 5). The 3-year OS rates were 97.0% versus 91.0% in the pertuzumab versus placebo arm ($\Delta$6.0%; 95% CI 0.08–12.1; $p = 0.0529$ [Supplementary Fig. 1]). The 5-year OS rates were 93.9% and 90.0% ($\Delta$3.9%; 95% CI 2.9–10.7; $p = 0.262$), respectively, with a positive trend towards the pertuzumab arm (HR 0.53; 95% CI 0.23–1.19 [Supplementary Fig. 1]).

### *Post hoc* exploratory subgroup analyses

Exploratory subgroup analyses demonstrated increased benefit of EFS and DFS in patients treated with pertuzumab versus placebo across prespecified subgroups, including disease stage and estrogen and progesterone receptor (ER- and PgR)-negative and -positive disease (Supplementary Fig. 2 and Supplementary Fig. 3).

Analysis of DFS according to tpCR status was performed. Data from both treatment arms combined showed a 3-year DFS rate of 93.4% (95% CI 88.6–98.1) in patients with tpCR versus 83.7% (95% CI 78.6–88.9) in those without; the 5-year DFS rate was 92.4% (95% CI 87.3–97.5) versus 76.9% (95% CI 71.0–82.8) (Fig. 3). When comparing the pertuzumab and placebo arms in patients with tpCR (Fig. 4), the 3-year DFS rate was 92.7% (95% CI 87.1–98.3) versus 95.7% (87.3–100); the 5-year DFS rate was 91.5% (95% CI 85.4–97.5) versus 95.7% (95% CI 87.3–100). In patients with residual invasive disease (Fig. 5), the 3-year DFS rate with pertuzumab versus placebo was 88.3% (95% CI 82.5–94.1) versus 76.8% (95% CI 67.4–86.2); the 5-year DFS rate was 82.1% (95% CI 75.2–89.1) versus 68.8% (95% CI 58.5–79.2).

### Prespecified biomarker analyses

Baseline biomarker values and prevalence were similar between the two treatment arms (Fig. 6). While a tpCR benefit with pertuzumab was observed among all biomarker subgroups, this was less apparent among patients with a *phosphatidylinositol-4,5-bisphosphate 3-kinase catalytic subunit alpha* (*PIK3CA*) mutation detected (Fig. 6). A slightly greater pertuzumab benefit versus trastuzumab was seen in patients with higher *HER2* mRNA levels than those with lower levels. Both observations appeared to be driven by an impact of the biomarker in the pertuzumab arm rather than an inverse or lack of impact in the placebo arm.

For EFS and DFS analyses, when both treatment arms were combined, patients with no *PIK3CA* mutation detected had better EFS (HR 0.45; 95% CI 0.27–0.75) and DFS (HR 0.48; 95% CI 0.28–0.81) rates than those with a *PIK3CA* mutation detected (Supplementary Fig. 4 and Supplementary Fig. 5). Prognostic trends were also observed for the HER2 immunohistochemistry

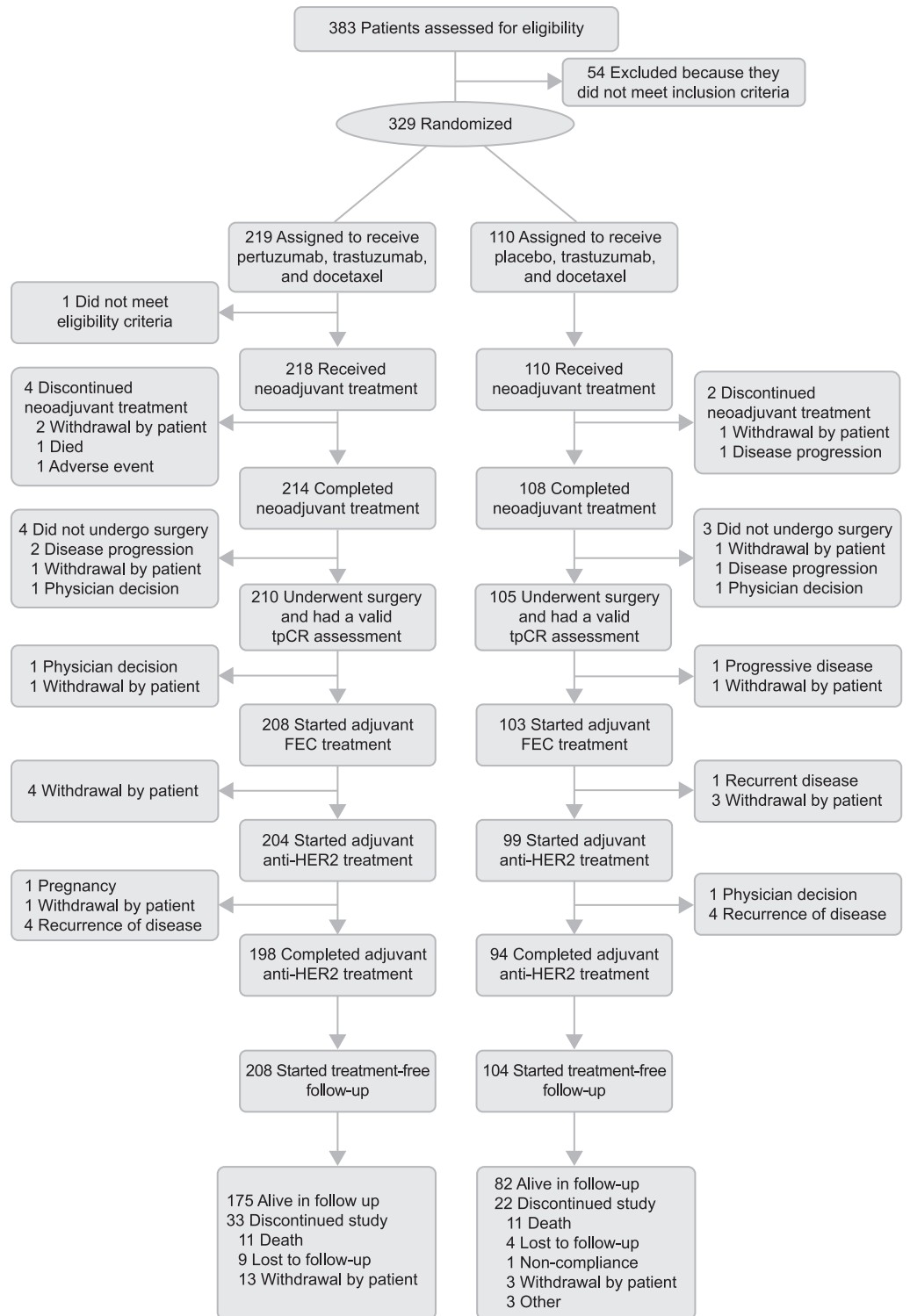

**Fig. 1 | CONSORT diagram.** FEC 5-fluorouracil, epirubicin, and cyclophosphamide. HER2 human epidermal growth factor receptor 2, tpCR total pathologic complete response.

(IHC) subgroups, with the HER2 IHC 3+ subgroup showing longer EFS (HR 0.70; 95% CI 0.41–1.20) and DFS (HR 0.78; 95% CI 0.44–1.38) compared with the HER2 IHC 1+/2+ subgroup.

## Safety
One patient in the pertuzumab arm discontinued from the study before receiving study treatment due to not meeting eligibility criteria,

resulting in 218/219 and 110/110 safety-evaluable patients in the pertuzumab and placebo arms, respectively.

A safety summary is presented in Table 2. During the neoadjuvant and adjuvant phases, 218 (100%) and 109 (99.1%) patients in the pertuzumab and placebo arms, respectively, had any-grade AEs. There was a similar incidence of serious (37 [17.0%] versus 15 [13.6%]) and grade ≥3 AEs (154 [70.6%] versus 75 [68.2%]) in the pertuzumab versus

**Table 1 | Site of first EFS event**

| Patients with events, *n* (%) | Pertuzumab group (*n* = 219) | Placebo group (*n* = 110) |
|---|---|---|
| Any EFS event | 32 (14.6) | 27 (24.5) |
| Locoregional progression before surgery | 2 (0.9) | 3 (2.7) |
| Locoregional recurrence after surgery | 2 (0.9) | 7 (6.4) |
| Distant metastasis without CNS metastases | 13 (5.9) | 12 (10.9) |
| CNS metastases | 6 (2.7) | 0 |
| Contralateral breast cancer | 4 (1.8) | 1 (0.9) |
| Second primary non-breast cancer | 2 (0.9) | 3 (2.7) |
| Death without previous event | 3 (1.4) | 1 (0.9) |

Patients who had an additional event within 60 days of their first event are reported in the category according to the following hierarchy: distant recurrence, locoregional recurrence, contralateral breast cancer, second primary non-breast cancer, and death without previous event.
*CNS* central nervous system, *EFS* event-free survival.

placebo arm. The number of patients who had grade ≥3 AEs in the pertuzumab versus placebo arm was 100 (48.1%) versus 45 (43.7%) in the 5-fluorouracil, epirubicin, and cyclophosphamide (FEC) treatment phase, and 23 (11.3%) versus 13 (13.1%) in the adjuvant anti-HER2 treatment phase. Grade ≥3 AEs that occurred during the overall neoadjuvant–adjuvant treatment period demonstrated an expected chemotherapy toxicity profile.

Any-grade diarrhea and upper respiratory tract infection were reported more frequently in the pertuzumab versus placebo arm. The three most common grade ≥3 AEs were neutropenia, leukopenia, and febrile neutropenia; serious AEs included febrile neutropenia, myelo-suppression, and pneumonia. Deaths due to AEs were few (two patients each in the pertuzumab arm [0.9%] and placebo arm [1.8%]). Significant left ventricular ejection fraction (LVEF) decline events were low and observed in two patients each in the pertuzumab (0.9%) and placebo (1.8%) arms (Table 2).

A total of 49 patients (22.8%) in the pertuzumab arm and 21 (19.3%) in the placebo arm with an LVEF ≥ 50% had a decrease from baseline of ≥10% ejection fraction points. A decrease from baseline of ≥10% ejection fraction points with an LVEF <50% was observed in two patients in each arm. No primary cardiac events (heart failure [New York Heart Association grade III or IV] and significant decline of LVEF) or secondary cardiac events occurred during any study periods.

## Discussion

In this 5-year follow-up of PEONY, long-term efficacy endpoints of EFS and DFS showed a clinically meaningful improvement with pertuzumab plus trastuzumab and docetaxel as neoadjuvant and adjuvant therapy, with favorable OS results (although the number of events was low in both arms).

Combinations of trastuzumab and pertuzumab have improved patient outcomes[6,9,10] over the years and are standard of care in HER2-positive breast cancer. In NeoSphere, progression-free survival and DFS at the 5-year follow-up suggested that neoadjuvant pertuzumab is beneficial when combined with trastuzumab and docetaxel[11]. Consistently, the EFS analysis in PEONY demonstrated a clinically meaningful long-term benefit with this combination. Although the long-term survival data are supportive of the primary endpoint in PEONY, those results are descriptive and should be interpreted with caution. In contrast to the NeoSphere trial, patients in the PEONY investigational arm received both pertuzumab and trastuzumab to complete 1 year of adjuvant HER2 treatment. A similar treatment strategy with the dual HER2 blockade continued for up to 1 year has also been evaluated in the APHINITY trial (NCT01358877) in the adjuvant setting, which demonstrated a 28% reduction in the risk of recurrence or death and an absolute invasive DFS benefit of 4.9% at 8 years in patients at high risk of recurrence (i.e. those with lymph node-positive disease)[7]. Exploratory subgroup analyses of PEONY

demonstrated increased EFS and DFS benefit in patients treated with pertuzumab versus placebo, across prespecified subgroups, including disease stage and hormone receptor-positive or -negative disease. The benefit was consistent across the majority of the patient subgroups and appeared to be more marked in patients with high-risk features (positive lymph node status and hormone receptor-negative disease). This is in line with the totality of the data and is a more attenuated position considering the exploratory nature of the analysis and the wide CIs in some of the subgroups (e.g. lymph node-negative).

As demonstrated in PEONY, several trials in the neoadjuvant setting have proved pCR to be a reliable surrogate parameter for long-term outcome of patients with HER2-positive disease[13–15]. Exploratory subgroup analyses were also conducted by surgery outcome and treatment arm; however, as the responder analysis does not preserve randomization, these results must be interpreted with caution. For patients with tpCR following previous anti-HER2 treatment, continuing after surgery with the same anti-HER2 (dual- or single-agent) therapeutic strategy did not appear to have any influence on DFS. Notably, the small number of DFS events (one in the placebo arm versus seven in the pertuzumab arm), and the overlapping 95% CIs between treatment arms for 5-year DFS, increase the uncertainty around these results. A separate pooled analysis of five studies with pertuzumab, trastuzumab, and che-motherapy showed that patient outcomes appear greatest for those who achieve a pCR and when the treatment includes pertuzumab and trastuzumab in both the neoadjuvant and adjuvant settings[16]. The totality of the data and current standard of care recommended by international guidelines for patients with HER2-positive early breast cancer at high risk of recurrence is 1 year of pertuzumab–trastuzumab therapy, regardless of the timing of surgery[17,18]. Although response-guided adjuvant treatment was not part of the PEONY design, the 3-year DFS rate of 88.3% in the PEONY dual anti-HER2 therapy arm was similar to the 3-year invasive DFS of 88.3% in the ado-trastuzumab emtansine (T-DM1) arm of the KATHERINE trial (NCT01772472)[19]. Based on the KATHERINE trial, 14 cycles of adjuvant T-DM1 have become the standard of care for patients with residual invasive disease[17–19]. However, most patients in KATHERINE received a minimum of six cycles of neoadjuvant therapy[19]. In the placebo arm of PEONY, all high-risk patients received trastuzumab without pertuzumab. A direct comparison of adjuvant T-DM1 with dual anti-HER2 therapy (pertuzumab plus trastuzumab) in patients who do not achieve a pCR is currently lacking, although ongoing phase III trials may help to elucidate the optimal adjuvant treatment strategy. Since some regions have no access to T-DM1 and some patients cannot complete treatment due to AEs, availability of effective adjuvant therapy containing pertuzumab plus trastuzumab will continue to be an important component of curative treatment in this setting.

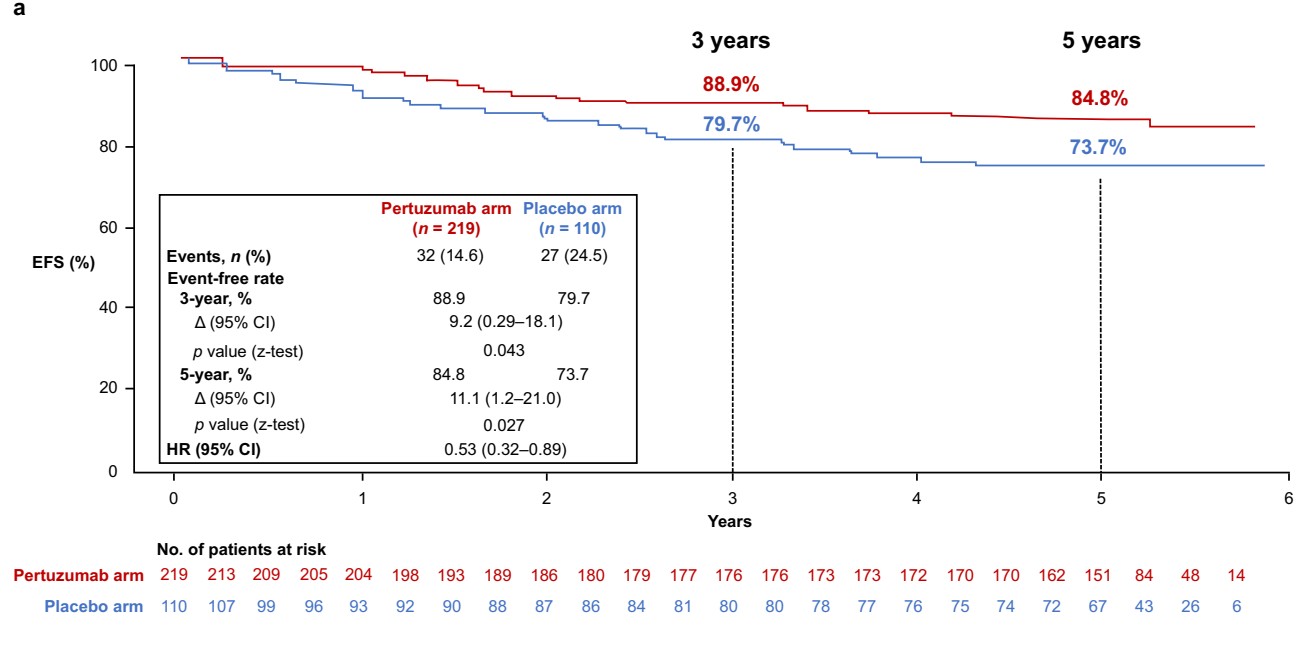

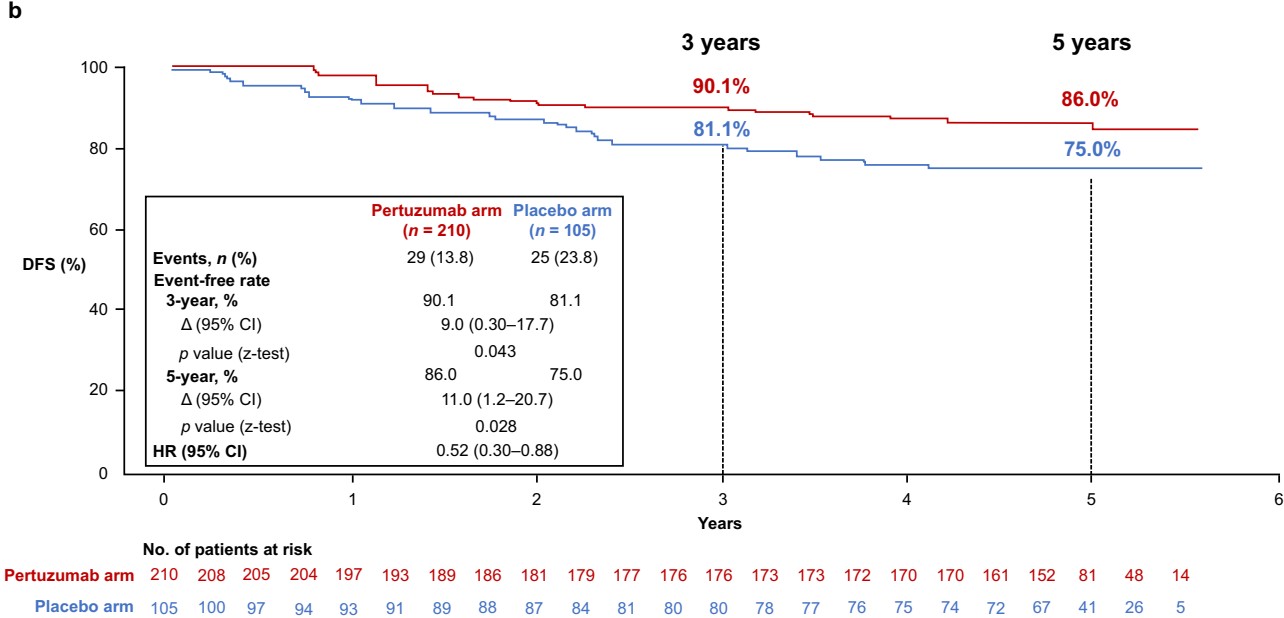

**Fig. 2 | Kaplan–Meier plots of the long-term efficacy endpoints EFS and DFS.** **a** EFS in the ITT population and **b** DFS in all patients who underwent surgery. 95% CIs and *p* values (two-sided) for differences in event rates were from z-tests using the standard errors for the Kaplan–Meier estimates. CI confidence interval, DFS disease-free survival, EFS event-free survival, HR hazard ratio, ITT intention-to-treat.

In the PEONY trial, most biomarker subgroups showed improved tpCR rates in the pertuzumab versus placebo arm, although this benefit seemed less clear in the subgroup of patients with *PIK3CA* mutations. In addition, a slightly greater benefit with pertuzumab versus trastuzumab was seen in the higher *HER2* mRNA subgroup compared with the lower *HER2* mRNA subgroup. Previous studies showed that tpCR rates are lower in patients with *PIK3CA* mutations compared with those without, independent of treatment arm[20–22]. A more pronounced benefit in patients with higher *HER2* mRNA compared with those with lower levels has been shown in various trials of HER2-positive breast cancer, independent of anti-HER2 therapy[20,21]. In the PEONY trial, the impact of these two biomarkers was observed in the pertuzumab arm, but not in the placebo arm, resulting in different magnitudes of benefit in the subgroups. Potential imbalances in prognostic factors in these subgroups may have played a role. The pooled analysis in PEONY showing a poorer long-term outcome for those with *PIK3CA* mutations and in the lower HER2 IHC subgroups was in line with PI3K/PTEN/AKT pathway alterations being linked to poorer outcomes in the APHINITY study (pooled arms)[23]. All biomarker results should be interpreted with caution due to the small sample sizes of the subgroups and the wide 95% CI ranges.

In PEONY, the incidence of AEs to monitor during the overall treatment period were consistent with the known safety profiles of the two regimens, with a higher incidence of diarrhea, rash, leukopenia (grade 3), leukopenia infection (mainly upper respiratory tract infection), and infusion-related reactions in the pertuzumab compared with the placebo arm. Overall, no new safety signals were identified in the study. The nature and severity of AEs reported were consistent with

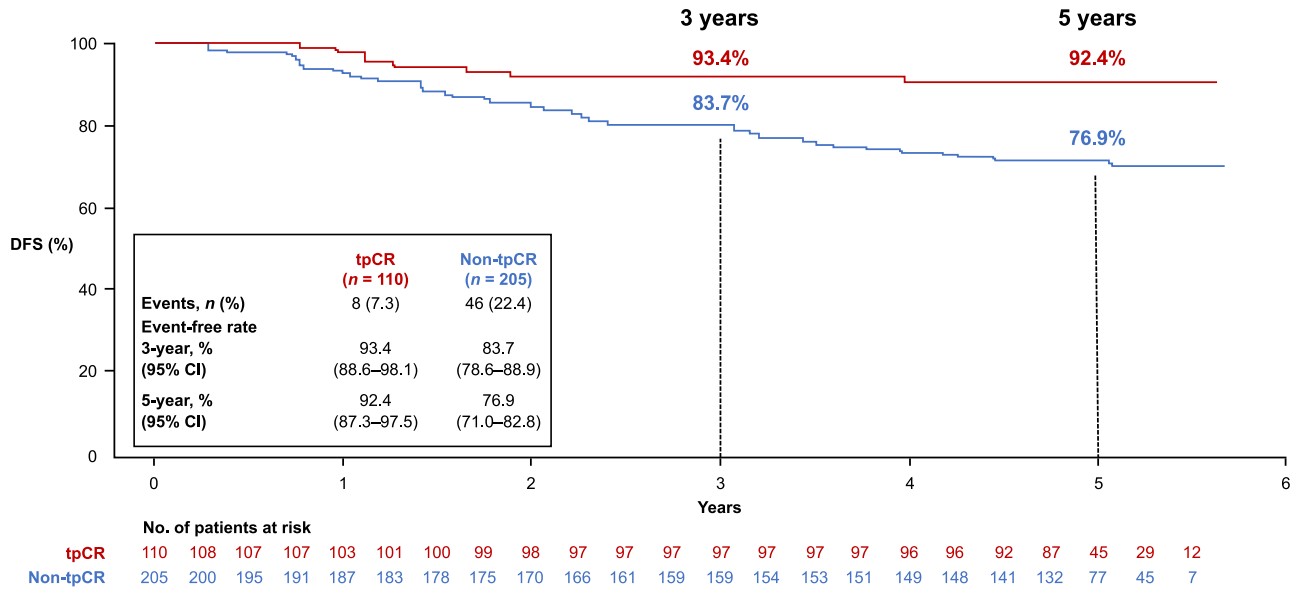

**Fig. 3 | Kaplan–Meier plot of DFS according to tpCR in the ITT population with both treatment arms combined.** CI confidence interval, DFS disease-free survival, ITT intention-to-treat, tpCR total pathologic complete response.

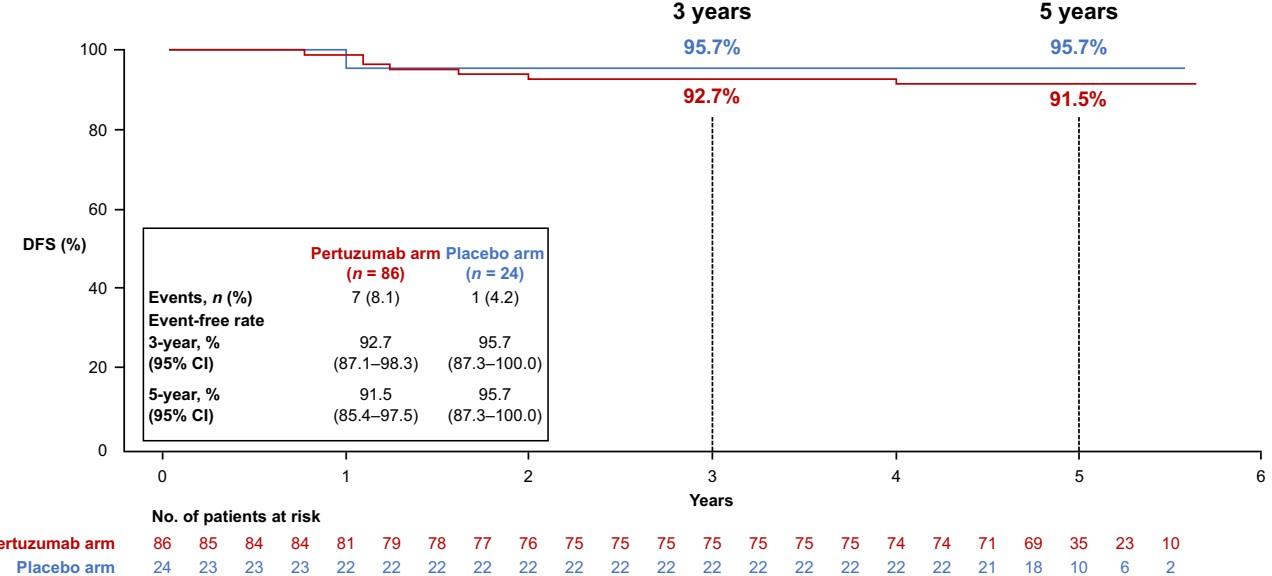

**Fig. 4 | Subgroup analysis of DFS in the tpCR population.** CI confidence interval, DFS disease-free survival, tpCR total pathologic complete response.

the known safety profile of pertuzumab, trastuzumab, and chemotherapy in the study population.

PEONY is a randomized, double-blind, placebo-controlled, phase III study that reports the long-term efficacy and safety of the addition of pertuzumab to trastuzumab in both the neoadjuvant and adjuvant setting in HER2-positive early breast cancer. One of the limitations of PEONY is that patients with residual invasive disease did not switch to receive standard-of-care T-DM1 treatment, as this option was not approved at the time the study was designed. A limitation of the present PEONY analyses is that the survival results, including EFS, DFS, and OS, are descriptive, as the study was not powered to detect differences in the ITT or any of the subgroups. In PEONY, the neoadjuvant treatment regimen included four cycles of a taxane-only regimen plus dual HER2 blockade, with the remaining three cycles of FEC administered post-surgery; whereas, currently, a common treatment strategy

encompasses the administration of the full course of chemotherapy before surgery for four to eight cycles[17]. Randomization was stratified by disease category (early stage or locally advanced) and hormone receptor status; however, a slightly higher proportion of patients in the pertuzumab arm had clinical lymph node-negative disease and primary tumor stage T2, which may have impacted the tpCR and EFS/DFS rates. The rates of cardiac toxicities were low in both arms; however, a limitation was that long-term follow-up of cardiac function was not followed in this study. Nonetheless, PEONY provides additional evidence for the use of pertuzumab and trastuzumab in combination with a taxane in the neoadjuvant setting, with tpCR rates similar to those seen in NeoSphere[10]. Dual HER2 blockade plus a taxane and carboplatin may allow for a non-anthracycline de-escalated chemotherapy approach[24]; while dual HER2 blockade plus a taxane has been shown to result in high pCR rates[25], it is as yet unclear whether

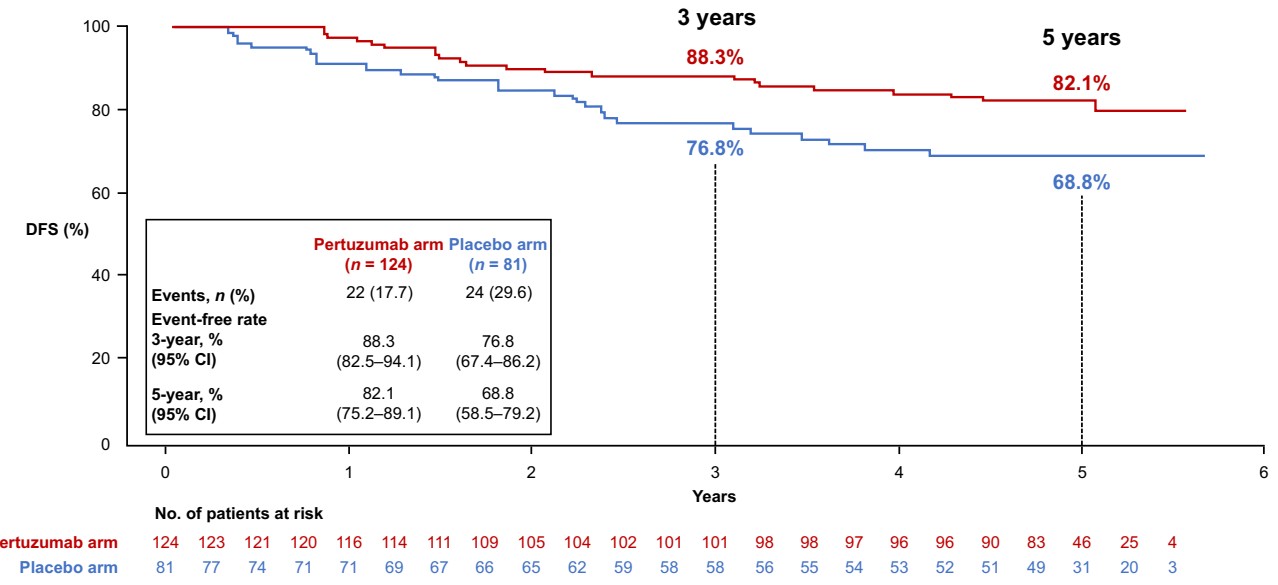

**Fig. 5 | Subgroup analysis of DFS in the population with residual invasive disease.** CI confidence interval, DFS disease-free survival.

| Biomarker | Total, n | Pertuzumab + trastuzumab + chemotherapy (n = 219) | | | Placebo + trastuzumab + chemotherapy (n = 110) | | | Odds ratio | 95% CI | Pertuzumab + trastuzumab + chemotherapy better | Placebo + trastuzumab + chemotherapy better |
| | | n | Responder | Response, % | n | Responder | Response, % | | | | |
|---|---|---|---|---|---|---|---|---|---|---|---|
| **All patients** | 329 | 219 | 86 | 39.3 | 110 | 24 | 21.8 | 0.43 | 0.25–0.73 | | |
| *HER2* mRNA | | | | | | | | | | | |
| ≤ median | 163 | 109 | 34 | 31.2 | 54 | 13 | 24.1 | 0.70 | 0.33–1.47 | | |
| > median | 161 | 107 | 51 | 47.7 | 54 | 10 | 18.5 | 0.25 | 0.11–0.55 | | |
| *HER3* mRNA | | | | | | | | | | | |
| ≤ median | 165 | 110 | 50 | 45.5 | 55 | 11 | 20.0 | 0.30 | 0.14–0.64 | | |
| > median | 161 | 107 | 35 | 32.7 | 54 | 12 | 22.2 | 0.59 | 0.28–1.25 | | |
| PTEN by H-score cytoplasma | | | | | | | | | | | |
| ≤ median | 165 | 110 | 45 | 40.9 | 55 | 14 | 25.5 | 0.49 | 0.24–1.01 | | |
| > median | 164 | 109 | 41 | 37.6 | 55 | 10 | 18.2 | 0.37 | 0.17–0.81 | | |
| PTEN by H-score nuclear | | | | | | | | | | | |
| ≤ median | 214 | 137 | 50 | 36.5 | 77 | 19 | 24.7 | 0.57 | 0.31–1.06 | | |
| > median | 115 | 82 | 36 | 43.9 | 33 | 5 | 15.2 | 0.23 | 0.08–0.65 | | |
| *PIK3CA* mutation status | | | | | | | | | | | |
| Mutated | 98 | 63 | 13 | 20.6 | 35 | 8 | 22.9 | 1.14 | 0.42–3.09 | | |
| Non-mutated | 230 | 155 | 73 | 47.1 | 75 | 16 | 21.3 | 0.30 | 0.16–0.58 | | |
| HER3 IHC membrane H-score | | | | | | | | | | | |
| High (> median) | 146 | 98 | 39 | 39.8 | 48 | 9 | 18.8 | 0.35 | 0.15–0.80 | | |
| Low (≤ median) | 174 | 115 | 44 | 38.3 | 59 | 12 | 20.3 | 0.41 | 0.20–0.86 | | |

1/100        1        100

**Fig. 6 | Biomarker subgroup analysis of tpCR rates in the ITT population.** Boxes represent sample sizes and odds ratios; whiskers represent 95% CI; the vertical dotted line represents the odds ratio for all patients. CI confidence interval, HER2 human epidermal growth factor receptor 2, IHC immunohistochemistry, ITT intention-to-treat; mRNA, messenger RNA, *PIK3CA phosphatidylinositol-4,5-bisphosphate 3-kinase catalytic subunit alpha*, PTEN phosphatase and tensin homolog, tpCR total pathologic complete response.

anthracyclines are required to optimize long-term survival outcomes in the absence of platinum therapy.

In patients with early-stage or locally advanced HER2-positive breast cancer, the magnitude of clinical benefit and the acceptable safety profile in PEONY confirms the positive benefit:risk ratio of dual pertuzumab–trastuzumab neoadjuvant/adjuvant therapy, adding to the totality of the evidence of its use in the early setting, irrespective of time of surgery.

## Methods
### Study oversight
The study was conducted in full accordance with the principles of the Declaration of Helsinki and the International Conference on Harmonisation E6 guideline for Good Clinical Practice. Approval for the protocol and amendments was obtained from an institutional review board and/or independent ethics committees (ethics committees belonged to specific participating sites/hospitals and are listed in Supplementary Table 6). All patients provided written informed consent.

### Study design and patients
PEONY was an Asia-Pacific, regional, multicenter, randomized, double-blind, placebo-controlled, phase III trial to evaluate treatment with pertuzumab, trastuzumab, and docetaxel (pertuzumab arm), compared with placebo, trastuzumab, and docetaxel (placebo arm). The study protocol is available with the previously published primary analysis (DOI: 10.1001/jamaoncol.2019.3692) and the study was registered at https://clinicaltrials.gov/ on October 23, 2015, prior to patient enrollment (ClinicalTrials.gov identifier: NCT02586025).

**Table 2 | Safety summary during the overall treatment period in the safety-evaluable population**

| Number of patients, n (%) | Pertuzumab group (n = 218) | Placebo group (n = 110) |
|---|---|---|
| Any-grade AEs | 218 (100) | 109 (99.1) |
| Ten most common AEs | | |
| Neutropenia | 154 (70.6) | 73 (66.4) |
| Leukopenia | 135 (61.9) | 67 (60.9) |
| Alopecia | 115 (52.8) | 56 (50.9) |
| Nausea | 84 (38.5) | 40 (36.4) |
| Anemia | 75 (34.4) | 37 (33.6) |
| Diarrhea | 89 (40.8) | 19 (17.3) |
| Alanine aminotransferase increased | 64 (29.4) | 41 (37.3) |
| Aspartate aminotransferase increased | 54 (24.8) | 34 (30.9) |
| Upper respiratory tract infections | 58 (26.6) | 14 (12.7) |
| Decreased appetite | 40 (18.3) | 13 (11.8) |
| Grade ≥3 AEs | 154 (70.6) | 75 (68.2) |
| During FEC phase | 100 (48.1) | 45 (43.7) |
| During adjuvant anti-HER2 phase | 23 (11.3) | 13 (13.1) |
| Neutropenia | 129 (59.2) | 61 (55.5) |
| Leukopenia | 75 (34.4) | 38 (34.5) |
| Febrile neutropenia | 11 (5.0) | 4 (3.6) |
| Anemia | 9 (4.1) | 5 (4.5) |
| Thrombocytopenia | 7 (3.2) | 1 (0.9) |
| Menstruation irregular | 8 (3.7) | 0 |
| Serious AEs | 37 (17.0) | 15 (13.6) |
| Febrile neutropenia | 9 (4.1) | 3 (2.7) |
| Pneumonia | 4 (1.8) | 1 (0.9) |
| Myelosuppression | 3 (1.4) | 0 |
| Grade 5 AEs[a] | 2 (0.9) | 2 (1.8) |
| AEs leading to treatment withdrawal | 2 (0.9) | 0 |
| AEs leading to treatment modification | 41 (18.8) | 17 (15.5) |
| AEs of special interest | 0 | 0 |
| Primary cardiac events (heart failure [NYHA functional classification III or IV] and significant LVEF decline[b]) | 0 | 0 |
| Significant LVEF decline events[b] | 2 (0.9) | 2 (1.8) |
| Secondary cardiac events | 0 | 0 |

*AE* adverse event, *FEC* 5-fluorouracil, epirubicin, and cyclophosphamide, *HER2* human epidermal growth factor receptor 2, *LVEF* left ventricular ejection fraction, *NYHA* New York Heart Association.

[a]Deaths due to an AE occurred in two patients (suicide in the neoadjuvant period and pneumonia in the treatment-free follow-up period) in the pertuzumab arm and two patients (sudden death in the treatment-free follow-up period) in the placebo arm.

[b]A significant LVEF decline is defined as an absolute decrease of at least 10 points below the baseline measurement and to an LVEF of <50%.

Patients were ≥18 years of age, had an Eastern Cooperative Oncology Group Performance Status of 0/1 and an LVEF of ≥55%, and were chemotherapy-naïve with a primary tumor size of >2 cm and histologically confirmed early-stage (T2–3, N0–1, M0) or locally advanced (T2–3, N2 or N3, M0; T4, any N, M0) HER2-positive breast cancer. The overall study design is presented in Supplementary Fig. 6.

Patients were enrolled from March 14, 2016, to March 13, 2017, and were randomized 2:1 using a web-based system and a permuted block randomization procedure to the pertuzumab or placebo arm. Randomization was stratified by disease category (early stage or locally advanced) and hormone receptor status (positive for ER and/or PgR, or negative for both). The investigators enrolled the patients, the site staff (unblinded) generated the randomization assignment tables that were entered into an interactive voice/web response system (IxRS); the block size was hidden in the randomization sequence. Patients were randomized via the IxRS system and assigned a treatment group and investigational product ID, while the true group information was blinded and kept in the system background. The investigators and the patients were blinded to the treatment assignment. All other individuals who were directly involved in the study remained blinded to the treatment assignment until completion of the primary analysis.

## Study procedures

Pertuzumab (PERJETA®, F. Hoffmann-La Roche Ltd [Basel, Switzerland]/Genentech, Inc. [South San Francisco, CA]; intravenous [IV]; 840 mg loading dose, followed by 420 mg maintenance doses), trastuzumab (Herceptin®, F. Hoffmann-La Roche Ltd/Genentech, Inc.; IV; 8 mg per kg loading dose followed by 6 mg per kg maintenance doses), and docetaxel (IV; 75 mg per m$^2$), or IV placebo, trastuzumab, and docetaxel, were administered for four cycles every 3 weeks in the neoadjuvant setting. For additional antibodies used in the study, see the Supplementary Information and reporting summary.

After completion, all eligible patients underwent surgery with their pathologic response evaluated. After surgery, patients received 500–600 mg per m$^2$ of 5-fluorouracil, 90–120 mg per m$^2$ of epirubicin, and 500–600 mg per m$^2$ of cyclophosphamide, every 3 weeks for three cycles (Cycles 5–7). Patients then continued HER2-targeted therapy in accordance with the initial randomization every 3 weeks until disease recurrence (as assessed by the investigator) or unacceptable toxicity, for up to 1 year.

For patients with tumors that were ER- and/or PgR-positive, hormonal agents were started at the end of FEC chemotherapy, with treatment planned for at least 5 years. Radiotherapy was given as clinically indicated at the end of FEC chemotherapy.

Additional to HER2, ER, and PgR testing, retrospective analyses of samples were performed for HER2 pathway-related biomarkers, including *HER2/HER3* mRNA, HER3, and PTEN protein expression, and *PIK3CA* mutations, before neoadjuvant treatment. *HER2/HER3* mRNA were analyzed using the cobas® z 480 analyzer, a quantitative reverse transcription polymerase chain reaction assay using protocols and processing instructions developed at Roche Molecular Diagnostics, Inc. (Pleasanton, CA). Expression levels were reported as a ratio of glucose-6-phosphate dehydrogenase expression. *PIK3CA* mutations in exons 1, 4, 7, 9, and 20 of DNA were detected using the commercially available cobas® *PIK3CA* Mutation Test (Roche Molecular Diagnostics, Inc. [Pleasanton, CA]). PTEN was stained using the anti-PTEN (clone 138G6) Rabbit mAb (Cell Signaling Technology® [Danvers, MA]) and HER3 IHC was evaluated by Ventana anti-HER3 (7.3.8) mouse monoclonal primary antibody on the Ventana Benchmark XT machine (Roche Tissue Diagnostics [Tucson, AZ]).

The samples from patients in mainland China were analyzed at the Roche Oncology Biomarker Development laboratory (Shanghai, China). The patient samples from the rest of the world (Korea, Taiwan, and Thailand) were examined at Targos Molecular Pathology GmbH (Kassel, Germany).

## Study endpoints

The primary efficacy outcome measure of PEONY was independent review committee-assessed tpCR (i.e. ypT0/is, ypN0 according to the current American Joint Committee on Cancer [AJCC] staging system). Here, we report the long-term efficacy endpoints of EFS, DFS, and OS, as well as safety.

EFS was defined as the time from randomization to first documentation of disease progression before surgery (determined according to the Response Evaluation Criteria In Solid Tumors [RECIST] v1.1, excluding contralateral disease in situ), disease recurrence after surgery (local, regional, distant, or contralateral, second primary non-breast cancer), or death by any cause. DFS was defined as

the time from surgery to the first documented disease recurrence after surgery (local, regional, distant, or contralateral, second primary non-breast cancer), or death by any cause. OS was defined as the time from randomization to death from any cause. Data from patients who had no event at the time of the analysis were censored as of the date they were last known to be alive and event-free. Severity of AEs was graded according to the National Cancer Institute's Common Terminology Criteria for AEs (NCI-CTCAE) v4.0[26].

Patients were followed for survival every 3 months (±28 days) for the first year and then every 6 months (±28 days) thereafter until disease progression/recurrence, or until 5 years after randomization of the last patient; whichever occurred first.

## Statistical analysis
The study aimed to randomize a total of 328 patients in a 2:1 ratio to the pertuzumab or placebo arm, respectively, to provide 85% power to detect an absolute increase in tpCR rate of 15% in the pertuzumab arm versus placebo arm at a two-sided significance level of 5%, assuming the tpCR rate was 20% in the placebo arm. PEONY was not powered to detect survival differences for secondary endpoints (EFS, DFS, OS). Results of these analyses are for descriptive purposes only. The ITT population comprised all randomized patients, whether or not the assigned study treatment was received. The safety-evaluable population consisted of all patients who received at least one dose of study drug.

EFS and OS were analyzed in the ITT population and DFS was analyzed for all patients who underwent surgery. Long-term outcomes were assessed by Kaplan–Meier methods, Cox proportional-hazards models, and a two-sided log-rank test (stratified by disease category [early stage or locally advanced] and hormone receptor status [positive for ER and/or PgR or negative for both]). The two-sided log-rank test was used to make an exploratory comparison of survival distribution between the two treatment arms. The Kaplan–Meier approach was used to estimate survival rates at various timepoints (i.e. 3 years, 5 years) for each treatment arm. The stratified Cox proportional-hazards model was used to estimate the HR between the two treatment arms (i.e., the magnitude of treatment effect) and its 95% CI.

Median values were used as cutoffs for each of the continuous biomarkers in the analysis of tpCR, except for *PIK3CA* mutations (*PIK3CA*-mutation-not-detected versus mutation-detected). For the biomarker subgroup analysis of tpCR rates, the logistic regression model was used to estimate the odds ratio and 95% CI (Wald test). For EFS and DFS, the two treatment arms were pooled to examine the prognostic effect of HER2 and *PIK3CA* given the small sample size of subgroups per treatment arm. All $p$ values were from two-sided tests and results were deemed to show a positive trend at $p < 0.05$. Analyses were conducted using SAS v9.4 (SAS Institute, Inc. [Cary, CA]).

## Reporting summary
Further information on research design is available in the Nature Portfolio Reporting Summary linked to this article.

## Data availability
The data supporting the findings of this study, including Source data, cannot be made available openly owing to their proprietary nature. Researchers may request access to individual patient-level data through the clinical study data request platform (https://vivli.org/ourmember/roche/). Requests to access the data from the trial can be submitted through: https://search.vivli.org/?search=NCT02586025. An independent review panel will review each data request based on the research proposal. On average it takes a few months to access data in the Vivli platform, but the timeline will vary depending on the number of data contributors, the number of studies, and the requester's availability to respond to comments. Data deposited in the Vivli

Standard Research Environment is free for the first 365 days, after which a daily charge will incur (see https://vivli.org/resources/requestdata/ for details). For up-to-date details on Roche's Global Policy on the Sharing of Clinical Information and how to request access to related clinical study documents, see here: https://go.roche.com/data_sharing. Anonymized records for individual patients across more than one data source external to Roche can not, and should not, be linked due to a potential increase in risk of patient re-identification. The study protocol is available with the previously published primary analysis (DOI: 10.1001/jamaoncol.2019.3692). The remaining data are available within the Article and its Supplementary Information.

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

## Acknowledgements

This research was funded by F. Hoffmann-La Roche Ltd (no grant number). F. Hoffmann-La Roche Ltd was involved in the study design, data interpretation, and decision to submit for publication in conjunction with the authors. Research support in the form of medical writing assistance, furnished by Sunaina Indermun, PhD, and Brian Law, PhD, of Nucleus Global, an Inizio company, was provided by F. Hoffmann-La Roche Ltd.

## Author contributions

Z.S., D.P., Y.W., and Z.J. conceived and designed the study. G.S. performed the statistical analysis, had full access to all the data in the study, and took responsibility for the integrity of the data and the accuracy of the data analysis. All authors (L.H., D.P., H.Y., W.L., S.W., S.C., N.L., Y.W., C.W., Y.-C.C., H-C.W., S.Y.K., J.H.S., K.S., S.L., Z.J., H.W., F.L., G.S., M.C., C.D., S.L.dH., E.R., Z.S.) contributed to the acquisition, analysis, or interpretation of the data. Z.S., L.H., H.W., G.S., C.D., S.L.dH., and M.C. drafted the manuscript. All authors contributed to the critical revision of the manuscript for important intellectual content. All authors approved the final manuscript and agree to be accountable for all aspects of the work.

## Competing interests

All authors report receiving research support in the form of third-party writing assistance for this manuscript, provided by F. Hoffmann-La Roche Ltd. L.H., D.P., H.Y., W.L., S.W., S.C., N.L., Y.W., C.W., Y.-C.C., H-C.W., S.Y.K., J.H.S., K.S., Z.J., and Z.S. report contracted research from F. Hoffmann-La Roche Ltd. S.L. reports contracted research from F. Hoffmann-La Roche Ltd/Genentech, Inc. via Roche Thailand Ltd. S.W. reports honoraria from F. Hoffmann-La Roche Ltd. H.W., C.D., S.L.dH., and E.R. report ownership interest (stocks, stock options, patent or other intellectual property or other ownership interest excluding diversified mutual funds) in F. Hoffmann-La Roche Ltd. F.L., M.C., S.L.dH., and E.R. report employment with F. Hoffmann-La Roche Ltd. H.W. reports employment with Roche Product Development. C.D. reports employment with Roche (China) Holding Ltd. G.S. reports employment with Hangzhou Tigermed Consulting Co., Ltd.

## Additional information

[1]Fudan University Shanghai Cancer Center, 200032 Shanghai, China. [2]Shanghai Medical College, Fudan University, 200032 Shanghai, China. [3]Harbin Medical University Cancer Hospital, 150040 Harbin, China. [4]Cancer Hospital of The University of Chinese Academy of Sciences, 310022 Hangzhou, China.

[5]The First Hospital of Jilin University, 130012 Changchun, China. [6]Sun Yat-sen University Cancer Center, 510060 Guangzhou, China. [7]Henan Cancer Hospital, 450003 Zhengzhou, China. [8]Guangdong General Hospital, 510060 Guangzhou, China. [9]Shandong Cancer Hospital, 250117 Jinan, China. [10]Fujian Medical University Union Hospital, 350001 Fuzhou, China. [11]Department of General Surgery, Mackay Memorial Hospital, 104 Taipei City, Taiwan. [12]Department of Surgery, China Medical University Hospital, 404 Taichung City, Taiwan. [13]Ajou University School of Medicine, 206 Suwon, Republic of Korea. [14]Korea University Guro Hospital, 08308 Seoul, Republic of Korea. [15]Ruijin Hospital, Shanghai Jiao Tong University School of Medicine, 200025 Shanghai, China. [16]Prince of Songkla University, 90110 Songkhla, Thailand. [17]The Affiliated Hospital of Military Medical Sciences (The 307th Hospital of Chinese. People's Liberation Army), 100071 Beijing, China. [18]Roche Product Development, 201203 Shanghai, China. [19]F. Hoffmann-La Roche Ltd, 4070 Basel, Switzerland. [20]Hangzhou Tigermed Consulting Co., Ltd, 310053 Shanghai, China. [21]Department of Translational Medicine Oncology, Roche (China) Holding Ltd, 201203 Shanghai, China. [22]Present address: Alentis Therapeutics AG, Allschwil, Switzerland. ✉e-mail: zhimingshao@fudan.edu.cn

