## [Peer Review File · Nature Communications]

Neoadjuvant–adjuvant pertuzumab in HER2-positive early breast cancer: Final analysis of the randomized phase III PEONY trialREVIEWER COMMENTS

Reviewer #1 (Remarks to the Author): with expertise in breast cancer, therapy

In this manuscript, Huang et al describe the long-term events (quantified as EFS and DFS rates) from the randomized PEONY trial, which investigated taxane/trastuzumab +/- pertuzumab  surgery  FEC  trastuzumab +/- pertuzumab. The primary endpoint of the trial, pCR, was previously reported. The authors find that EFS and DFS are both significantly improved with addition of pertuzumab. While other randomized trials (NeoSphere and APHINITY) have confirmed the degree of benefit of adding pertuzumab in the neoadjuvant or adjuvant setting, the PEONY trial is unique in that it evaluated addition of pertuzumab in both the neoadjuvant AND adjuvant setting.

I found this manuscript to be a generally well written and clear report on EFS/DFS in the PEONY population, with some interesting additional discussion about biomarkers (largely confirmatory of prior findings). I have the following comments that I feel may help to improve the manuscript:

- In the intro first paragraph, I would focus more on APHINITY than on CLEOPATRA since this is a paper about early stage breast cancer and therefore APHINITY is more germane
- It is a bit unusual that both DFS and EFS are listed alongside each other. I would recommend reminding the reader in the results section text what is the difference between EFS and DFS definitions i.e. what type of events made up the small difference between these figures
- Since this trial was randomized in the neoadjuvant AND adjuvant setting, and no pts got adjuvant T-DM1 even if they had residual disease it seems, these authors are in a unique position to estimate the benefit of adjuvant pertuzumab both for pts WITH pCR after HP, and for pts WITHOUT pCR after HP. This is a question that comes up clinically quite a lot. I know the Swain paper is referenced and does suggest benefit of P post-pCR, but to do a formal analysis on this here would be very interesting and also clinically meaningful.
- In the final 2 paragraphs of Results, within the safety section, I am a bit confused. The sentence "Deaths due to AEs and significant LVEF decline events..." to me reads like the AEs were DUE to LVEF decline, but I don't think that is what the authors mean. Please clarify.

Also in the next paragraph, the sentence "No primary cardiac events...occurred" confuses me, because isn't that contradicted by the LVEF declines enumerated in the paragraph above?

Reviewer #2 (Remarks to the Author): with expertise in breast cancer, therapy

This manuscript reports the long term follow up of PEONY, a phase 3 RCT that evaluated the use of pertuzumab in the neoadjuvant/adjuvant settings for HER2+ disease. Other studies have evaluated the use of pertuzumab in the curative setting but this study design is unique because Neosphere and TRYPHAENA evaluated pertuzumab only in the neoadjuvant setting and APHINITY evaluated pertuzumab only in the adjuvant setting. PEONY is the only study to evaluate a full year of pertuzumab given in both neoadjuvant and adjuvant settings. Significant improvements in tpCR, EFS and DFS were observed with pertuzumab. These data are important because they are the first to support the use of pertuzumab in both settings which is a common clinical practice that has not been supported by clinical trial evidence to date. The paper is well written. However it would be strengthened by addressing a number of issues.

MAJOR

In the Discussion section, the authors state, "If the initial risk is high, such as in patients with node-positive disease, receiving combined dual anti-HER2 neoadjuvant therapy and achieving a pCR should not trigger treatment de-escalation to trastuzumab monotherapy." This conclusion/statement is not data-driven and should be removed or rephrased as this study has not proven that the use of pertuzumab in the adjuvant setting improves EFS/DFS in the setting of pCR. To demonstrate adjuvant pertuzumab benefits pts in setting of high risk disease and pCR, pts with pCR would need to be randomized to pertuzumab vs no pertuzumab.

On a related note, the authors only present an analysis of DFS based on tpCR for treatment arms combined. The DFS analysis should also be shown for those who had tpCR and for those with residual disease based on receipt of pertuzumab (i.e. show the DFS KM curve for

patients with pCR treated with placebo vs treated with pertuzumab). Is pertuzumab mainly benefitting those who had residual disease? Add KM figures and add to Suppl Fig 2 forest plot.

Discussion section: The authors should acknowledge that an unanswered question is whether patients with residual disease do better with continuing HP or switching to T-DM1, which is a standard therapy used in many parts of the world after non-pCR based on KATHERINE.

Discussion section: The authors should acknowledge the benefit of pertuzumab seems to be restricted to LN+ as per suppl figs 2 and 3. This is a similar finding from APHINITY

Discussion section it states, "In the PEONY trial, most biomarker subgroups showed improved prognosis in the pertuzumab versus placebo arm." This statement is vague and confusing. Prognostic information is of little use clinically. A more important question is whether these biomarkers are predictive of benefit with pertuzumab. Data should be clearly presented in the results section, specifically did patients with PIK3CA mut, low PTEN, low HER2, and/or high HER3 appear to benefit less from pertuzumab?

Based on forest plot, it seems pertuzumab does not benefit those with PIK3CA mutation or low HER2? These data should be presented in the text in the Results section and should be acknowledged/discussed as other studies have also indicated a dampening of benefit with dual HER2-targeted therapies in the setting of PIK3CA mutations (for example, in CLEOPATRA).

Table 2: All grade AEs are provided, the authors need to show grade 3/4 events separately in table.

A higher proportion of pts were in the pertuzumab arm with node negative disease and T2 disease (vs T3). Given this may impact tpCR and EFS/DFS rates, it should be discussed.

The authors report low rates of cardiac toxicity in both arms. However, it should be acknowledged in the discussion section (when discussing limitations of the study) that long

term cardiac function was not followed in this study. In the BCIRG 006 study, LVEF was followed for 10 years to pick up subclinical cardiac dysfunction and results indicate that anthracyclines reduce cardiac function long term. Cardiac toxicity for anthracycline-treated patients may be subclinical and may not emerge in first year of receiving therapy.

MINOR:

Baseline demographics and breast conserving surgery rates are in the study design section rather than the results section.

Suppl fig 2 and 3 could be shortened by removing the analysis for ER+PR-, ER-PR+, ER-PR-, ER+PR+ as the numbers are too small in several groups and the data is captured above in the ER-PR- vs ER and/or PR+ section

Report how many patients received growth factor during chemo phase.

Instead of calling (PIK3CA)-no-mutation-detected subgroup, change to PIK3CA-wild type

In safety section it states "Patients who had grade ≥ 3 AEs in the pertuzumab versus placebo arm were 100 (48.1%) versus 45 (43.7%), and 23 (11.3%) versus 13 204 (13.1%), respectively." What is the 23 and 13 referring to? Grade 4 events? Need to state clearly.

COMMENTS FOR THE AUTHORS

Final analysis of the PEO 1 NY trial of neoadjuvant–adjuvant pertuzumab in HER2-positive early breast cancer

My review will focus on the statistical aspects because of my background.

Purpose and design:

This paper reports final long-term efficacy and safety analysis results for the randomized, multicenter, double-blind, placebo-controlled, phase III PEONY trial (NCT02586025).

In this trial, Patients (n = 329) were randomized at 2:1 ratio to receive neoadjuvant pertuzumab/placebo with trastuzumab and docetaxel, followed by adjuvant 5-fluorouracil, epirubicin, and cyclophosphamide, then pertuzumab/placebo with trastuzumab for up to 1 year. Randomization was stratified by disease category (early-stage or locally advanced) and hormone receptor status (positive for ER and/or PgR, or negative for both).

The primary efficacy outcome measure of PEONY was independent review committee³⁴² assessed tpCR (i.e. ypT0/is, ypN0 according to the current American Joint Committee on Cancer [AJCC] staging system), but this paper long-term efficacy endpoints of EFS, DFS, and OS, as well as safety. It is noted that this study (PEONY) was not powered to detect survival differences for secondary endpoints (EFS, DFS, OS).

Trial registration: NCT02586025

Statistical methods:

EFS and OS were analyzed in the ITT population and DFS was analyzed for all patients who underwent surgery.

The Kaplan–Meier approach was used to estimate survival rates at various timepoints (i.e. 3 years, 5 years) for each treatment arm.

The two-sided log-rank test (stratified by disease category and hormone receptor status) was used to make an exploratory comparison of survival distribution between the two treatment arms. The stratified Cox proportional hazards model was used to estimate the HR between the two treatment arms (i.e., the magnitude of treatment effect) and its 95% CI.

For the biomarker subgroup analysis of tpCR rates, the logistic regression model was used to estimate the odds ratio and 95% CI (Wald test). For EFS and DFS, the two treatment arms were pooled to examine the prognostic effect of HER2 and PIK3CA.

Results/findings

They concluded that:

- Five-year event-free survival estimates are 84.8% with pertuzumab and 73.7% with

- placebo (hazard ratio 0.53; 95% confidence interval 0.32–0.89; $p = 0.014$);
- 5-year disease free survival rates are 86.0% and 75.0%, respectively (hazard ratio 0.52; 95% confidence interval 0.30–0.88; $p = 0.014$).
- Safety data are consistent with the known pertuzumab safety profile and generally comparable between arms, except for diarrhea (40.8% in pertuzumab vs 17.3% in placebo).

PEONY confirms the positive benefit:risk ratio of neoadjuvant/adjuvant pertuzumab, trastuzumab, and docetaxel treatment in this patient population.

Overall Comments:

The paper is overall very well-written: the design part is clear, the analyses are appropriate and clearly described, especially with what was described in the protocol. This review has no major comments on the manuscript. My comments are below for the authors to consider.

Specific Comments

I think the statistical analysis section (lines 369-381) could be re-organized. Specifically,

- The sentence “Data from patients who had no event at the time of the analysis were censored as of the date they were last known to be alive and event-free.” (lines 369-371) could move to the “study endpoints” section, because this is related to definition of the endpoints (OS, DFS, etc).
- Lines 372-274 could move up to line 360, just before “EFS and OS were ...”.

Final analysis of the PEONY trial of neoadjuvant–adjuvant pertuzumab in HER2-positive early breast cancer

Reviewer 1 In this manuscript, Huang et al describe the long-term events (quantified as EFS and DFS rates) from the randomized PEONY trial, which investigated taxane/trastuzumab +/- pertuzumab  surgery  FEC  trastuzumab +/- pertuzumab. The primary endpoint of the trial, pCR, was previously reported. The authors find that EFS and DFS are both significantly improved with addition of pertuzumab. While other randomized trials (NeoSphere and APHINITY) have confirmed the degree of benefit of adding pertuzumab in the neoadjuvant or adjuvant setting, the PEONY trial is unique in that it evaluated addition of pertuzumab in both the neoadjuvant AND adjuvant setting. I found this manuscript to be a generally well written and clear report on EFS/DFS in the PEONY population, with some interesting additional discussion about biomarkers (largely confirmatory of prior findings). I have the following comments that I feel may help to improve the manuscript:	
Reviewer 1 comments	Author responses
1. In the intro first paragraph, I would focus more on APHINITY than on CLEOPATRA since this is a paper about early stage breast cancer and therefore APHINITY is more germane	We have amended the Introduction on page 5 to focus on HER2-positive early breast cancer and APHINITY: ‘The availability of human epidermal growth factor receptor 2 (HER2)-targeted therapy has dramatically changed the prognosis for patients with HER2-positive early breast cancer. Trastuzumab added to chemotherapy has been associated with improved pathologic complete response (pCR) rates and long-term survival¹⁻³. However, the majority of patients do not achieve a pCR^{1,4} and of these, approximately one third will develop disease recurrence at 3 years⁴. Dual HER2 blockade with pertuzumab and trastuzumab has proven more effective than single-agent trastuzumab in the treatment of HER2-positive early breast cancer. The APHINITY study (NCT01358877) showed that addition of pertuzumab to standard adjuvant therapy significantly improves invasive disease-free survival (DFS)⁵⁻⁷.’
2. It is a bit unusual that both DFS and EFS are listed alongside each other. I would recommend reminding the reader in the results section	Please note the efficacy outcome measures included both EFS and DFS. The definitions for EFS and DFS are provided in the Methods section on

text what is the difference between EFS and DFS definitions i.e. what type of events made up the small difference between these figures	page 16, and Table 1 lists 'locoregional progression before surgery' as an event under the term EFS. We have added "Post-surgery" to the description of DFS in the Results section on page 7.
3. Since this trial was randomized in the neoadjuvant AND adjuvant setting, and no pts got adjuvant T-DM1 even if they had residual disease it seems, these authors are in a unique position to estimate the benefit of adjuvant pertuzumab both for pts WITH pCR after HP, and for pts WITHOUT pCR after HP. This is a question that comes up clinically quite a lot. I know the Swain paper is referenced and does suggest benefit of P post-pCR, but to do a formal analysis on this here would be very interesting and also clinically meaningful.	Prof. Swain analyzed patient-level data from five randomized trials evaluating trastuzumab, pertuzumab, or both, as part of systemic neoadjuvant and adjuvant therapy for HER2-positive early breast cancer (ref 16 in the manuscript; DOI: 10.3390/cancers14205051). Regardless of pCR status, after adjusting for baseline factors, a greater reduction in the risk of an EFS event was seen in patients administered pertuzumab plus trastuzumab in both settings versus those administered only trastuzumab in both settings, or pertuzumab plus trastuzumab in the neoadjuvant setting and trastuzumab only in the adjuvant setting. Please also see our response to Reviewer 2 Comment 2 for more information on pertuzumab for pCR versus non-pCR.
4. In the final 2 paragraphs of Results, within the safety section, I am a bit confused. The sentence "Deaths due to AEs and significant LVEF decline events..." to me reads like the AEs were DUE to LVEF decline, but I don't think that is what the authors mean. Please clarify. Also in the next paragraph, the sentence "No primary cardiac events...occurred" confuses me, because isn't that contradicted by the LVEF declines enumerated in the paragraph above?	Thank you for flagging this – we have amended the text on page 9 for clarity: 'Deaths due to AEs were few (two patients each in the pertuzumab arm [0.9%] and placebo arm [1.8%]). Significant left ventricular ejection fraction (LVEF) decline events were low and observed in two patients each in the pertuzumab (0.9%) and placebo (1.8%) arms (Table 2).' A primary cardiac event is defined as heart failure (NYHA III or NYHA IV) and a drop in LVEF of at least 10 ejection fraction points from baseline and to below 50%. Two patients in each group showed significant decrease in LVEF, but these patients were not diagnosed with heart failure (NYHA III or NYHA IV) at the same time. According to this definition of a primary cardiac event, there were no occurrences in this study.

Reviewer 2	
This manuscript reports the long term follow up of PEONY, a phase 3 RCT that evaluated the use of pertuzumab in the neoadjuvant/adjuvant settings for HER2+ disease. Other studies have evaluated the use of pertuzumab in the curative setting but this study design is unique because Neosphere and TRYPHAENA evaluated pertuzumab only in the neoadjuvant setting and APHINITY evaluated pertuzumab only in the adjuvant setting. PEONY is the only study to evaluate a full year of pertuzumab given in both neoadjuvant and adjuvant settings. Significant improvements in tpCR, EFS and DFS were observed with pertuzumab. These data are important because they are the first to support the use of pertuzumab in both settings which is a common clinical practice that has not been supported by clinical trial evidence to date. The paper is well written. However it would be strengthened by addressing a number of issues.	
Reviewer 2 comments	Author responses
Major	
1. In the Discussion section, the authors state, "If the initial risk is high, such as in patients with node-positive disease, receiving combined dual anti-HER2 neoadjuvant therapy and achieving a pCR should not trigger treatment de-escalation to trastuzumab monotherapy." This conclusion/statement is not data-driven and should be removed or rephrased as this study has not proven that the use of pertuzumab in the adjuvant setting improves EFS/DFS in the setting of pCR. To demonstrate adjuvant pertuzumab benefits pts in setting of high risk disease and pCR, pts with pCR would need to be randomized to pertuzumab vs no pertuzumab.	We have removed this statement from the Discussion section.
2. On a related note, the authors only present an analysis of DFS based on tpCR for treatment arms combined. The DFS analysis should also be shown for those who had tpCR and for those with residual disease based on receipt of pertuzumab (i.e. show the DFS KM curve for patients with pCR treated with placebo vs treated with pertuzumab). Is pertuzumab mainly benefitting those who had residual disease? Add KM figures and add to Suppl Fig 2 forest plot.	We have added this content to the Results section (page 8), added the KM plots to the Supplementary section (new Supplementary Figs. 4 & 5) and updated the forest plot (Supplementary Fig. 3). 'Analysis of DFS according to tpCR status was performed. Data from both treatment arms combined showed a 3-year DFS rate of 93.4% (95% CI 88.6–98.1) in patients with tpCR versus 83.7% (95% CI 78.6–88.9) in those without; the 5-year DFS rate was 92.4% (95% CI 87.3–97.5) versus 76.9% (95% CI 71.0–82.8) (Fig. 3). When comparing the pertuzumab and placebo arms in patients with tpCR (Supplementary Fig. 4), the 3-year DFS rate was 92.7% (95% CI 87.1–98.3) versus 95.7% (87.3–100); the 5-year DFS rate was 91.5% (95% CI 85.4–97.5) versus 95.7% (95% CI 87.3–100). In patients with residual invasive disease

	(Supplementary Fig. 5), the 3-year DFS rate with pertuzumab versus placebo was 88.3% (95% CI 82.5–94.1) versus 76.8% (95% CI 67.4–86.2); the 5-year DFS rate was 82.1% (95% CI 75.2–89.1) versus 68.8% (95% CI 58.5–79.2).’
3. Discussion section: The authors should acknowledge that an unanswered question is whether patients with residual disease do better with continuing HP or switching to T-DM1, which is a standard therapy used in many parts of the world after non-pCR based on KATHERINE.	We have added some new text to the Discussion section (page 11–12): ‘Exploratory subgroup analyses were also conducted by surgery outcome and treatment arm; however, as the responder analysis does not preserve randomization, these results must be interpreted with caution. For patients with tpCR following previous anti-HER2 treatment, continuing after surgery with the same anti-HER2 (dual- or single-agent) therapeutic strategy did not appear to have any influence on DFS. Notably, the small number of DFS events (one in the placebo arm versus seven in the pertuzumab arm), and the overlapping 95% CIs between treatment arms for 5-year DFS, increase the uncertainty around these results. A separate pooled analysis of five studies with pertuzumab, trastuzumab, and chemotherapy showed that patient outcomes appear greatest for those who achieve a pCR and when the treatment includes pertuzumab and trastuzumab in both the neoadjuvant and adjuvant settings¹⁶. The totality of the data and current standard of care recommended by international guidelines for patients with HER2-positive early breast cancer at high risk of recurrence is 1 year of pertuzumab–trastuzumab therapy, regardless of the timing of surgery^{17,18}. Although response-guided adjuvant treatment was not part of the PEONY design, the 3-year DFS rate of 88.3% in the PEONY dual anti-HER2 therapy arm was similar to the 3-year invasive DFS of 88.3% in the ado-trastuzumab emtansine (T-DM1) arm of the KATHERINE trial (NCT01772472)¹⁹. Based on the KATHERINE trial, 14 cycles of adjuvant T-DM1 have become the standard of care for patients with residual invasive disease¹⁷⁻¹⁹. However, most patients in KATHERINE received a minimum of six cycles of neoadjuvant therapy¹⁹.

	In the placebo arm of PEONY, all high-risk patients received trastuzumab without pertuzumab. A direct comparison of adjuvant T-DM1 with dual anti-HER2 therapy (pertuzumab plus trastuzumab) in patients who do not achieve a pCR is currently lacking, although ongoing phase III trials may help to elucidate the optimal adjuvant treatment strategy. Since some regions have no access to T-DM1 and some patients cannot complete treatment due to AEs, availability of effective adjuvant therapy containing pertuzumab plus trastuzumab will continue to be an important component of curative treatment in this setting.'
4. Discussion section: The authors should acknowledge the benefit of pertuzumab seems to be restricted to LN+ as per suppl figs 2 and 3. This is a similar finding from APHINITY	We have added some new text to the Discussion section (page 10): 'A similar treatment strategy with the dual HER2 blockade continued for up to 1 year has also been evaluated in the APHINITY trial (NCT01358877) in the adjuvant setting, which demonstrated a 28% reduction in the risk of recurrence or death and an absolute invasive DFS benefit of 4.9% at 8 years in patients at high risk of recurrence (i.e. those with lymph node-positive disease)⁷. Exploratory subgroup analyses of PEONY demonstrated increased EFS and DFS benefit in patients treated with pertuzumab versus placebo, across prespecified subgroups including disease stage and hormone receptor-positive or -negative disease. Benefit with addition of pertuzumab was also seen in patients with lymph node-positive disease.'
5. Discussion section it states, "In the PEONY trial, most biomarker subgroups showed improved prognosis in the pertuzumab versus placebo arm." This statement is vague and confusing. Prognostic information is of little use clinically. A more important question is whether these biomarkers are predictive of benefit with pertuzumab. Data should be clearly presented in the results section, specifically did patients with PIK3CA mut, low PTEN, low HER2, and/or high HER3 appear to benefit less from pertuzumab? Based on forest plot, it seems pertuzumab does not benefit those with PIK3CA mutation or low HER2? These data should be presented	The predictive benefit is reflected by the analysis shown in the forest plot in Fig. 4, which highlights that the majority of biomarker subgroups show pertuzumab benefit independent of biomarker status. The main subgroup where the HR seems less beneficial is in the PIK3CA-mutated subgroup (HR 1.14) as highlighted by the reviewer. This result seems driven by a different pCR rate in patients with PIK3CA-mutated breast cancer versus those with no mutation detected in the pertuzumab arm, while this was not observed in the placebo arm, and therefore the comparison between the arms seems different per subgroup.

in the text in the Results section and should be acknowledged/discussed as other studies have also indicated a dampening of benefit with dual HER2-targeted therapies in the setting of PIK3CA mutations (for example, in CLEOPATRA).

In prior studies, including NeoSphere, a higher pCR rate for those without a *PIK3CA* mutation has been shown for patients receiving HER2-targeted therapy with trastuzumab, pertuzumab, or both. Therefore, the benefit of pertuzumab has been shown to be independent of *PIK3CA* mutation status. The results in the pertuzumab arm of the PEONY trial were in line with the NeoSphere findings, but the placebo arm in PEONY did not show these results. We would like to highlight that the CLEOPATRA trial in metastatic breast cancer also showed pertuzumab benefit independent of the mutation status (impact of the mutation present in both arms). The impact of the *PIK3CA* mutation has mainly been shown to be independent of HER2 treatment arm so far. The current finding in PEONY may be different due to some imbalances in other prognostic variables in the smaller-sized trastuzumab arm. For the *HER2* mRNA subgroups, the benefit is seen in both the high and low subgroups, but the magnitude of benefit differs. Also, the better pCR rate in the higher *HER2* mRNA subgroup in the pertuzumab arm is in line with prior findings from NeoSphere, KRISTINE and TRYPHAENA. The inverse association of the HER2 level and tpCR in the trastuzumab arm seems to influence the magnitude of benefit.

For EFS and DFS analyses, the arms were pooled due to the small number of events in subgroups per treatment arm and the focus was therefore on prognostic value analysis. These data showed that those with *PIK3CA* mutations and lower HER2 expression had worse outcomes in the overall pooled population, in line with the findings from APHINITY in which PI3K/PTEN/AKT pathway alterations and lower HER2 copy number have also been linked to poorer outcome.

In the Results section of the biomarker analyses, we have edited the text on the *PIK3CA* mutation subgroup and focused more on the comparison of arms on page 8–9

'While a tpCR benefit with pertuzumab was observed among all biomarker subgroups, this was less apparent among patients with a phosphatidylinositol-4,5-bisphosphate 3-kinase catalytic subunit alpha (*PIK3CA*) mutation detected (Fig. 4). A slightly greater pertuzumab benefit versus trastuzumab was seen in patients with higher *HER2* mRNA levels than those with lower levels. Both observations appeared to be driven by an impact of the biomarker in the pertuzumab arm rather than an inverse or lack of impact in the placebo arm. For EFS and DFS analyses, when both treatment arms were combined, patients with no *PIK3CA* mutation detected had better EFS (HR 0.45; 95% CI 0.27–0.75) and DFS (HR 0.48; 95% CI 0.28–0.81) rates than those with a *PIK3CA* mutation detected (Supplementary Fig. 6 and Supplementary Fig. 7). Prognostic trends were also observed for the *HER2* immunohistochemistry (IHC) subgroups, with the *HER2* IHC 3+ subgroup showing longer EFS (HR 0.70; 95% CI 0.41–1.20) and DFS (HR 0.78; 95% CI 0.44–1.38) compared with the *HER2* IHC 1+/2+ subgroup.'

We have also added additional text to the Discussion section on page 12.

'In the PEONY trial, most biomarker subgroups showed improved tpCR rates in the pertuzumab versus placebo arm, although this benefit seemed less clear in the *PIK3CA*-mutated subgroup. In addition, a slightly greater benefit with pertuzumab versus trastuzumab was seen in the higher *HER2* mRNA subgroup compared with the lower *HER2* mRNA subgroup. Previous studies showed that tpCR rates are lower in patients with *PIK3CA* mutations compared with those without, independent of treatment arm²¹⁻²³. A more pronounced benefit in patients with higher *HER2* mRNA compared with those with lower levels has been shown in various trials of *HER2*-positive breast cancer, independent of *HER2* therapy^{21,22}. In the PEONY trial, the impact of these two biomarkers was observed in the pertuzumab arm, but not in the placebo arm, resulting in different magnitudes of benefit in the subgroups. Potential imbalances in prognostic factors in these

	subgroups may have played a role. The pooled analysis in PEONY showing a poorer long-term outcome for those with PIK3CA mutations and in the lower HER2 IHC subgroups was in line with PI3K/PTEN/AKT pathway alterations being linked to poorer outcomes in the APHINITY study (pooled arms)²⁴. All biomarker results should be interpreted with caution due to the small sample sizes of the subgroups and the wide 95% CI ranges.' We have also removed the p-values for the exploratory subgroup analyses.
6. Table 2: All grade AEs are provided, the authors need to show grade 3/4 events separately in table.	Please note that grade ≥ 3 AEs are shown separately to the all-grade AEs, in rows 13–21 of Table 2.
7. A higher proportion of pts were in the pertuzumab arm with node negative disease and T2 disease (vs T3). Given this may impact tpCR and EFS/DFS rates, it should be discussed.	We have added this to the Discussion section (page 13): 'Randomization was stratified by disease category (early stage or locally advanced) and hormone-receptor status; however, a slightly higher proportion of patients in the pertuzumab arm had clinical lymph node-negative disease and primary tumor stage T2, which may have impacted the tpCR and EFS/DFS rates.'
8. The authors report low rates of cardiac toxicity in both arms. However, it should be acknowledged in the discussion section (when discussing limitations of the study) that long term cardiac function was not followed in this study. In the BCIRG 006 study, LVEF was followed for 10 years to pick up subclinical cardiac dysfunction and results indicate that anthracyclines reduce cardiac function long term. Cardiac toxicity for anthracycline-treated patients may be subclinical and may not emerge in first year of receiving therapy.	We have added this to the Discussion section (page 13): 'The rates of cardiac toxicities were low in both arms; however, a limitation was that long-term follow-up of cardiac function was not followed in this study.'
Minor	
9. Baseline demographics and breast conserving surgery rates are in the study design section rather than the results section.	We have corrected the 'Study design' subheading within the Results section (page 6) to read 'Patient population' instead.

10. Suppl fig 2 and 3 could be shortened by removing the analysis for ER+PR-, ER-PR+, ER-PR-, ER+PR+ as the numbers are too small in several groups and the data is captured above in the ER-PR- vs ER and/or PR+ section	We have amended Supplementary Figs. 2 & 3 accordingly.
11. Report how many patients received growth factor during chemo phase.	In Fig. 1, 214 patients in the pertuzumab arm and 108 patients in the placebo arm completed neoadjuvant chemotherapy and neoadjuvant anti-HER2 treatment. After adjuvant FEC treatment, 198 patients in the pertuzumab arm and 94 patients in the placebo arm completed adjuvant anti-HER2 treatment.
12. Instead of calling (PIK3CA)-no-mutation-detected subgroup, change to PIK3CA-wild type	As the cobas[®] assay was developed for detecting one of the 20 hotspot PIK3CA mutations and not the absence of a mutation (wild type), we feel it is more appropriate to refer to 'no-mutation detected'.
13. In safety section it states "Patients who had grade ≥ 3 AEs in the pertuzumab versus placebo arm were 100 (48.1%) versus 45 (43.7%), and 23 (11.3%) versus 13 204 (13.1%), respectively." What is the 23 and 13 referring to? Grade 4 events? Need to state clearly.	The 100 versus 45 and 23 versus 13 data were referring to the incidence of grade ≥ 3 AEs in the pertuzumab versus placebo arms in the FEC treatment phase and in the anti-HER2 treatment phase, respectively. We have reworded the sentence on page 9 for clarity: 'The number of patients who had grade ≥ 3 AEs in the pertuzumab versus placebo arm was 100 (48.1%) versus 45 (43.7%) in the 5-fluorouracil, epirubicin, and cyclophosphamide (FEC) treatment phase, and 23 (11.3%) versus 13 (13.1%) in the adjuvant anti-HER2 treatment phase.'

Reviewer 3 The paper is overall very well-written: the design part is clear, the analyses are appropriate and clearly described, especially with what was described in the protocol. This review has no major comments on the manuscript. My comments are below for the authors to consider.	
Reviewer 3 comments	Author responses
I think the statistical analysis section (lines 369-381) could be re-organized. Specifically,  1. The sentence "Data from patients who had no event at the time of the analysis were censored as of the date they were last known to be alive and event-free." (lines 369-371) could move to the "study 	 1. This has been moved to the study endpoints section on page 16 2. We have moved these sentences as requested

endpoints” section, because this is related to definition of the endpoints (OS, DFS, etc).	
--	--

- | | |
|--|--|
| 2. Lines 372-274 could move up to line 360, just before “EFS and OS were ...”. | |
|--|--|

Additional editorial requests:

- Nature editorial policy checklist: we will submit this with the revised manuscript
- Nature reporting summary: we will submit this with the revised manuscript
- CONSORT checklist: we will submit this with the revised manuscript

REVIEWERS' COMMENTS

Reviewer #1 (Remarks to the Author):

The authors have responded satisfactorily to my comments.

Reviewer #2 (Remarks to the Author):

most of this reviewer's comments have been addressed sufficiently with the exception of two:

1. Suppl figures 4/5 should be part of the main paper. Although the numbers are small for patients with tpCR, it is important to show from both a patient perspective and healthcare economics perspective and should be a result that is easy to see.

2. The authors' response is not adequate to the following comment: "The authors should acknowledge the benefit of pertuzumab seems to be restricted to LN+ as per suppl figs 2 and 3. This is a similar finding from APHINITY"

In their response ("Exploratory subgroup analyses of PEONY demonstrated increased EFS and DFS benefit in patients treated with pertuzumab versus placebo, across prespecified subgroups including disease stage and hormone receptor-positive or -negative disease. Benefit with addition of pertuzumab was also seen in patients with lymph node-positive disease," the authors dance around the fact that there is no clear evidence of benefit with pertuzumab in node negative disease. This section should be revised to clearly acknowledge this. For example, "benefit with the addition of pertuzumab appears to be restricted to/only observed in those with node positive disease"

Reviewer #3 (Remarks to the Author):

The authors have addressed my questions / comments very well in the revision and I have no further questions. Thanks!

Final analysis of the PEONY trial of neoadjuvant–adjuvant pertuzumab in HER2-positive early breast cancer

Reviewer 1 The authors have responded satisfactorily to my comments.	
Reviewer 2 Most of this reviewer's comments have been addressed sufficiently with the exception of two:	
Reviewer 2 comments	Author responses
1. Suppl figures 4/5 should be part of the main paper. Although the numbers are small for patients with tpCR, it is important to show from both a patient perspective and healthcare economics perspective and should be a result that is easy to see.	We have moved Supplementary Fig. 4 & 5 to the main body of the paper – these are now labelled as Fig. 4 & 5.
2. The authors' response is not adequate to the following comment: "The authors should acknowledge the benefit of pertuzumab seems to be restricted to LN+ as per suppl figs 2 and 3. This is a similar finding from APHINITY" In their response ("Exploratory subgroup analyses of PEONY demonstrated increased EFS and DFS benefit in patients treated with pertuzumab versus placebo, across prespecified subgroups including disease stage and hormone receptor-positive or -negative disease. Benefit with addition of pertuzumab was also seen in patients with lymph node-positive disease," the authors dance around the fact that there is no clear evidence of benefit with pertuzumab in node negative disease. This section should be revised to clearly acknowledge this. For example, "benefit with the addition of pertuzumab appears to be restricted to/only observed in those with node positive disease"	We thank the reviewer for the comment and acknowledge the reviewer's opinion. Please consider the following revision of this part: "The benefit was consistent across the majority of the patient subgroups and appeared to be more marked in patients with high-risk features (positive lymph node status and hormone receptor-negative disease). This is in line with the totality of the data and is a more attenuated position considering the exploratory nature of the analysis and the wide CIs in some of the subgroups (e.g. lymph node negative)." The rationale for the revision is as follows:  1. The lymph node-negative patient subgroup is quite small, comprising approximately 80 patients (25%) and a total of 5 events, despite the point estimate of hazard ratio (HR) favoring the pertuzumab arm. Given the limited sample size and event count, it is difficult to draw any meaningful conclusions for this specific subgroup in the PEONY study. For that reason, the

	discussion in the manuscript is restricted to the lymph node-positive subgroup. 2. From the study design perspective, APHINITY was designed for the adjuvant setting and used lymph node status (positive/negative) as one of the pre-defined stratification factors. From the APHINITY study design and statistical perspective, we can state that: the node positive (N+) cohort continues to derive clear IDFS benefit from the addition of P: HR 0.72 (95% CI 0.60-0.87). The benefit in terms of 8-year IDFS is 4.9% [86.1% vs 81.2%]. In the N-cohort, the IDFS HR is 1.01 with >92% of patients being event-free in both arms at 8 years (https://doi.org/10.1016/j.annonc.2022.06.009). While the PEONY study was designed for the neoadjuvant followed by adjuvant settings, the stratification factors include disease stage and hormone receptor status. The outcome of pertuzumab–trastuzumab in patients with residual disease is an exploratory endpoint, furthermore, as the responder analysis does not preserve randomization, these results must be interpreted with caution.
--	---

Reviewer 3

The authors have addressed my questions / comments very well in the revision and I have no further questions. Thanks!